# mTOR-Mediated Autophagy Regulates Cadmium-Induced Kidney Injury via Pyroptosis

**DOI:** 10.3390/ijms26062589

**Published:** 2025-03-13

**Authors:** Yuan Hu, Kui Wang, Jie Xu, Guohuan Wan, Yiyi Zhao, Yajing Chen, Kangfeng Jiang, Xiaobing Li

**Affiliations:** College of Veterinary Medicine, Yunnan Agricultural University, Kunming 650201, China; 17783722977@163.com (Y.H.); chris_wong001@yeah.net (K.W.); xujie2454@163.com (J.X.); m18419436036@163.com (G.W.); zhaoyiyi913@163.com (Y.Z.); yajingchen426@163.com (Y.C.)

**Keywords:** cadmium, kidney injury, mTOR, NLRP3, pyroptosis, autophagy

## Abstract

The heavy metal cadmium (Cd) affects the global livestock production economy mainly through the contamination of feed raw materials and secondary contamination in feed processing, and it also poses a serious threat to food safety and human health. The nucleotide-binding oligomerization domain-like pyrin-domain-containing protein 3 (NLRP3) inflammasome is a key regulatory element of pyroptosis, which is engaged in kidney injury. Meanwhile, autophagy is also involved in renal inflammation. Mammalian target of rapamycin (mTOR) plays an important role in pyroptosis and autophagy, but its function in Cd-induced kidney injury remains unclear. In this study, we explored the role of mTOR-mediated autophagy and pyroptosis in kidney injury caused by Cd exposure and elucidated its underlying mechanism. Our data showed that Cd exposure reduced the integrity of kidney cell membranes, increased the expression of pyroptosis-associated proteins, and promoted the release of inflammatory cytokines. Subsequently, a notable attenuation in Cd-induced pyroptosis was observed following the administration of CY-09, an NLRP3 inhibitor. In addition, Cd exposure promoted autophagy in kidney cells. Importantly, in both in vivo and in vitro experiments, rapamycin, an mTOR inhibitor, downregulated the expression of pyroptosis-related proteins, thereby significantly improving Cd-induced kidney injury. In summary, our results indicate that mTOR-mediated autophagy has a significant protective effect on NLRP3 inflammasome-dependent kidney injury induced by Cd exposure, thus providing new insights into the prevention and treatment of Cd poisoning.

## 1. Introduction

Cadmium (Cd) is a highly toxic metal known to cause serious health problems in humans and animals [1]. There is wide distribution of Cd in the atmosphere, water, soil, and food, which means that most people are inevitably exposed to low levels of cadmium for long periods [2,3,4,5]. It is accumulated in the human body for a long time through the food chain, with a half-life of about 25–30 years, and about 50% of Cd accumulates in the kidneys, leading to serious damage to the body [6,7]. In the 21st century, Cd has been exploited and abused by humans and has become a dangerous toxicant that pollutes the environment, posing a great challenge to the earth’s environment, the ecological balance, and the health of animals and humans [8,9]. The feeding patterns of grazing animals facilitate direct contact with the external environment, and feed contamination also allows cadmium to easily accumulate in the actual production of the animals [10,11]. Long-term exposure to cadmium can lead to various health issues, including anemia, urinary calcium loss, and damage to the kidneys and liver, and severe cadmium accumulation can ultimately result in animal death and impaired animal performance [12]. Cd, a pervasive renal toxin in the environment, is absorbed by the digestive tract and transported to the kidneys via the circulatory system [13]. Subsequently, the substance is filtered by the glomerulus and reabsorbed by the renal tubules. Following reabsorption, dissociated free Cd ions can further bind to and decompose metallothionein produced by the renal tubular epithelial cells, which causes severe damage to the renal tubular cells and leads to renal dysfunction [14]. For example, it can increase the incidence of kidney stones and reduce the glomerular filtration rate [15,16,17,18].

The mammalian target of rapamycin (mTOR) signaling pathway plays a pivotal role in the regulation of several fundamental cellular processes, including cell growth, proliferation, survival, and metabolism [19,20]. mTOR is a serine threonine protein kinase consisting of the catalytic subunits of two large multiprotein complexes: mTOR complex 1 (mTORC1) and mTOR complex 2 (mTORC2) [21]. Previous studies have demonstrated that mTORC1 and mTORC2 are significantly transformed in renal cell populations in vivo, driving the pathogenesis of kidney disease [22]. The use of mTOR inhibitors has been found to ameliorate chronic renal failure [23,24,25], whereas overactivation of mTOR can lead to proteinuria, neutrophil damage, and renal tubular cell injury, which can affect renal function [26]. Autophagy, also referred to as the autophagic lysosomal pathway, denotes the formation of autophagosomes through the engulfment of cytoplasmic components, organelles, and proteins destined for degradation within the cell. The formation of autophagic lysosomes occurs when autophagosomes fuse with lysosomes, and this process involves the degradation of damaged protein contents, which is essential for maintaining dynamic intracellular homeostasis. In addition, it plays a crucial role in cell survival and maintenance [27,28]. Importantly, the initiation, progression, and termination of autophagy are finely regulated by the mTOR signaling pathway [29]. In a study using Drosophila renal cells, inhibition of mTOR signaling resulted in an increase in the spacing of the autophagic septum [30].

Pyroptosis, also known as inflammatory necroptosis, is characterized by cell swelling but with an intact nucleus, DNA fragmentation, the formation of pores in the plasma membrane, and the release of inflammatory contents. Studies have confirmed that the activation of the mTOR signaling pathway leads to the activation of the nucleotide-binding oligomerization domain (NOD)-like receptor (NLR) family, which contains a hemoprotein domain 3 (NLRP3), and then the occurrence of pyroptosis [31,32]. The NLRP3 inflammasome activation is a major part of the classical activation of pyroptosis, resulting in the activation of pro-caspase-1 as well as the cleavage of gasdermin D (GSDMD) along with the release of cytosolic cytokines (*IL-1β* and *IL-18*), which lead to intense inflammation and cell death [33,34,35,36]. However, the mechanism through which mTOR activation in response to Cd exposure influences the autophagy process as well as the potential for Cd exposure to exacerbate inflammatory responses by affecting pyroptosis, leading to renal damage, has not yet been comprehensively evaluated.

In this study, we explored the effects of Cd on kidney damage in mice and cell damage in bovine kidney cells. In addition, we investigated the underlying mechanisms using specific agonists and inhibitors. This study may provide strong evidence for the molecular mechanism of Cd-induced kidney injury and the regulatory role of autophagy in inflammation.

## 2. Results

### 2.1. Cd Exposure Induces Pyroptosis in MDBK Cells

In this study, MDBK cells were employed as an in vitro model to investigate the nephrotoxic effects of Cd. The cells were exposed to different concentrations of Cd (0, 10, 20, 40, and 80 μM) for 12 h. As illustrated in Figure 1A,B, the findings indicated that the cell viability of MDBK cells diminished with increasing Cd concentrations, as evidenced by the CCK-8 assay. Moreover, the LDH content in the MDBK cells increased in a dose-dependent manner. Meanwhile, immunofluorescence analysis also showed colocalization of PI influx with the cell nucleus (Figure 1C), suggesting that high concentrations of Cd led to the destruction of the cell membrane’s integrity. Transmission electron microscopy further showed that cells exposed to 20 μM Cd exhibited cell swelling and vacuolar degeneration (Figure 1D). Moreover, Cd exposure significantly increased the levels of *IL-1β* and *IL-18* in MDBK cells compared with the control group (Figure 1E). ROS levels in MDBK cells after Cd stimulation were detected by flow cytometry, and it was subsequently found that ROS increased with increasing cadmium concentrations (Figure 1F). Notably, stimulation with Cd led to a notable upregulation of pyroptosis-associated proteins (NLRP3, C-caspase-1, N-GSDMD) in comparison with the control group (Figure 1G). Taken together, Cd exposure can induce pyroptosis in MDBK cells.

### 2.2. Inhibition of NLRP3 Alleviates Cd-Induced Pyroptosis in MDBK Cells

In this study, we employed CY-09, an inhibitor of NLRP3, to elucidate the role of the NLRP3 inflammasome in Cd-induced nephrotoxicity. The immunofluorescence results demonstrated that the reduction in cell membrane integrity by Cd stimulation was alleviated by CY-09 pretreatment (Figure 2A). The expression levels of *IL-1β* and *IL-18* in MDBK cells were also markedly reduced by CY-09 (Figure 2B). Furthermore, CY-09 pretreatment significantly reduced the protein levels of the pyroptosis-related genes (NLRP3, C-caspase-1, N-GSDMD) that were markedly upregulated by Cd (Figure 2C). In addition, our findings showed a notable reduction in mTOR phosphorylation level following CY-09 pretreatment (Figure 2D), suggesting a possible interaction between NLRP3 and mTOR. These findings indicate Cd induces pyroptosis in an NLRP3 inflammasome-independent manner.

### 2.3. Cd Exposure Promotes Autophagy in MDBK Cells

Under enhanced conditions of external stimuli, autophagy can promote cell survival, depending on the type of cellular stress. The transmission electron microscopy revealed the presence of autophagic vesicles in cells exposed to 20 μM Cd (Figure 3A), indicating Cd exposure promoted autophagy in MDBK cells. Additionally, the autophagosome expression was quantified using the lysosomal inhibitor chloroquine (CQ), and the results showed that CQ resulted in aggregation of LC3B in Cd-treated cells (Figure 3B,). As shown in Figure 3C, mRNA expression of autophagy-related genes (*ATG5*, *LC3B*) induced by Cd exposure was increased compared with the NC group, while the expression of the autophagy substrate *P62* decreased. Furthermore, the protein levels of autophagy-associated proteins (LC3B, ATG5) were observed to increase progressively with increasing Cd concentration, whereas the protein level of the autophagy substrate, P62, was significantly decreased when compared with the control group (Figure 3D). Based on these findings, Cd stimulation promotes autophagy levels in MDBK cells.

### 2.4. Activation of mTOR Exacerbates Cd-Induced Pyroptosis

Autophagy is a vital process for maintaining cellular homeostasis and is essential for cellular survival. Nevertheless, excessive autophagy can ultimately lead to cell death. In the present study, the mTOR agonist MHY1485 was employed to investigate the regulatory mechanisms of autophagy in Cd-induced kidney injury. As shown in Figure 4A, pretreatment with MHY1485 resulted in diminished expression of the autophagy-related proteins ATG5 and LC3B, as well as augmented levels of the autophagy substrate P62, in addition to an increase in the mTOR phosphorylation level (Figure 4B). Additionally, the autophagosome was quantified using the lysosomal inhibitor chloroquine (CQ), and the results showed that CQ resulted in aggregation of LC3B in Cd-treated cells, which was significantly reduced following MHY1485 pretreatment (Figure 4C). This suggests that Cd-induced autophagy is blocked by mTOR activation. Notably, the immunofluorescence results showed that pretreatment with MHY1485 further enhanced Cd-stimulated PI influx, demonstrating that cell membrane integrity was further reduced following autophagy blocked (Figure 4D). Moreover, pretreatment with MHY1485 further increased the expression of *IL-1β* and *IL-18* (Figure 4E) and pyroptosis-related proteins (NLRP3, C-caspase-1, N-GSDMD) in Cd-exposed MDBK cells (Figure 4F). The results of this study indicate that activation of mTOR exerts an inhibitory effect on autophagy, which exacerbates Cd-induced renal cell pyroptosis, suggesting a protective role for autophagy in Cd-induced kidney injury.

### 2.5. Inhibition of mTOR Attenuates Cd-Induced Pyroptosis in MDBK Cells

In addition, we further investigated the role of autophagy in Cd-induced kidney injury with the use of the mTOR inhibitor RAPA. As shown in Figure 5A, pretreatment with RAPA upregulated the expression of the autophagy-related proteins ATG5 and LC3B, inhibited the level of the autophagy substrate P62, as well as decreased the phosphorylation level of mTOR (Figure 5B). Additionally, the autophagosome was detected using the lysosomal inhibitor CQ and showed that RAPA pretreatment significantly increased autophagic flux (Figure 5C). As illustrated in Figure 5D, RAPA pretreatment resulted in a reduction in Cd-induced PI influx and an increase in cell membrane integrity. Furthermore, the expression of the pyroptosis-related inflammatory factors *IL-1β* and *IL-18* decreased significantly (Figure 5E), and the protein levels of pyroptosis-related proteins (NLRP3, C-caspase-1, N-GSDMD) were also found to be significantly reduced (Figure 5F). These findings indicate that the activation of autophagy by the inhibition of mTOR expression mitigated the Cd-induced pyroptosis-associated renal cell injury.

### 2.6. Inhibition of mTOR Alleviates Cd-Induced Kidney Injury in Mice

As shown in Figure 6A, the weight gain of mice in the Cd poisoning group was markedly diminished compared with the control group. However, the weight gain of mice in the mTOR inhibitor RAPA-treated group was significantly higher than that of mice in the Cd-intoxicated group. There was no significant difference in the kidney weights of the three groups of mice (Figure 6B). Furthermore, the CREA and BUN contents of the Cd-intoxicated group were significantly higher than those of the control group, while the CREA and BUN contents of the RAPA-treated group were considerably lower in comparison with those of the Cd-intoxicated group (Figure 6C,D), suggesting that RAPA can alleviate the renal dysfunction caused by Cd in mice. Similarly, the pathological changes observed in the kidney tissue of mice in the Cd-intoxicated group, including glomerular hypertrophy, hyperplasia, thickening of the basement membrane of the renal tubules, and narrowing of the tubular lumen, were mitigated by the intervention of RAPA (Figure 6E). Detection of ROS activity in the kidney tissues of mice in each group revealed that the fluorescence intensity of ROS in the Cd-intoxicated group was significantly higher than that in the control group, whereas the ROS activity of mice in the RAPA-treated group was significantly lower than that in the Cd-intoxicated group (Figure 6F). The transmission electron microscopy results showed that autophagosome morphology could be observed in both the Cd-intoxicated group and the RAPA-treated group (Figure 6G). Furthermore, the expression of autophagy-related proteins was significantly increased after RAPA intervention (Figure 6H). Meanwhile, the secretion of *IL-1β* and *IL-18* in the Cd-intoxicated group was also inhibited after RAPA intervention (Figure 6I), whereas the expression of pyroptosis-related proteins induced by Cd exposure was significantly reduced by RAPA (Figure 6J). These results demonstrate that mTOR inhibition suppresses pyroptosis through the activation of autophagy, which in turn attenuates Cd exposure-induced renal inflammatory injury in mice.

## 3. Discussion

The most important and direct factor in animal food safety is feed pollution. Heavy metal Cd mainly affects the global animal husbandry production economy through feed raw material pollution and secondary pollution in feed processing, and it also seriously endangers food safety and human health [37]. Cd can be absorbed by the human body through the gastrointestinal tract, respiratory tract, and skin [38]. The complexes formed accumulate and reabsorb in the target organ kidneys, and Cd induces pyroptosis through the ROS/NLRP3/caspase-1 pathway in the body, thus having a combined effect on nephrotoxicity [39]. Previous studies have shown that Cd exposure leads to excessive ROS production, which promotes pyroptosis in NRK-52E cells [40]. Cd has been demonstrated to act as a potent inducer of oxidative stress. However, excessive stress can also cause tissue damage by inducing pyroptosis [41]. The administration of Cd to rats prior to the induction of nephrotoxicity has been demonstrated to reduce oxidative stress by modifying biochemical functions, thereby mitigating the damage to kidney tissue [42]. So far, most of the research on excessive nephrotoxicity has focused on apoptosis, oxidative stress, and necrosis. However, the role of the pyroptosis-inducing cascade of inflammation in Cd-induced nephrotoxicity has not been elucidated. Therefore, in this study, we mainly investigated the damage caused by the inflammation caused by pyroptosis activated by Cd exposure in kidney cells and kidney tissue. This experiment also showed that Cd exposure induced pyroptosis in MDBK cells and promoted the gene expression of inflammatory factors (*IL-1β*, *IL-18*) in MDBK cells, and similar results were observed in in vivo experiments.

Pyroptosis is known as a new programmed pro-inflammatory cell death, which occurs mainly due to NLRP3 inflammasome activation, followed by triggering and amplifying the inflammatory response through the activation of pro-caspase-1 and GSDMD; pro-inflammatory cytokines are cleaved to their mature form [31,43]. Cd exposure induces pyroptosis of carp anterior kidney lymphocytes by activating NLRP3, and the inhibition of NLRP3 activity can reduce the extent of lymphocyte pyroptosis [44]. However, the mechanism of NLRP3 inflammasome and pyroptosis in Cd-induced kidney and renal cell injury has not been directly reported. In this study, the results showed that Cd contamination activated the NLRP3 inflammasome in mouse kidneys and MDBK cells, resulting in increased expressions of caspase-1 and GSDMD. Previous studies have shown synergistic effects on inflammation and cytotoxicity by activating NLRP3 inflammasomes, inducing necroptosis and autophagic cell death in NRK-52E cells; cinnamaldehyde inhibits NLRP3 inflammasome activation, resulting in reduced albuminuria and reduced glomerulosclerotic kidney inflammation in mice [39,45,46]. Therefore, in this experiment, we used CY-09, a targeted inhibitor of NLPR3, to determine the role of NLRP3 in Cd exposure-induced MDBK injury. The results showed that inhibition of NLRP3 inflammasome activity could inhibit the expression of pyroptosis-related proteins, gene expression of inflammatory factors, and cell membrane integrity in bovine kidney cells. These results suggest that NLRP3 inflammasome-dependent pyroptosis plays a role in Cd-induced cytotoxicity in MDBK.

Autophagy is a type of cell that removes damaged or harmful components through catabolism and recycling, maintaining a dynamic balance of nutrients and energy, and it is also an important protective mechanism [47]. Dysregulation of autophagy leads to acute kidney injury, incomplete renal repair after acute kidney injury, and chronic kidney disease of various etiologies; targeting the autophagy pathway may have a therapeutic effect in the treatment of kidney disease [48,49]. Autophagy mainly involves ATG5 and LC3B and related regulators of the autophagy substrate P62 [50]. In this experiment, we found that Cd exposure increased the expression of ATG5 and LC3B proteins and decreased the autophagic substrate P62, further suggesting that Cd stimulation promoted autophagy levels in MDBK cells. The removal of the NLRP3 inflammasome activator via autophagy has been demonstrated to reduce inflammasome activation and inflammation. Furthermore, autophagy has been shown to play a protective role in certain inflammatory diseases associated with the NLRP3 inflammasome [51,52,53]. Based on these findings, we conducted further research into the interaction between autophagy and the major pyroptosis pathway activated by the NLRP3 inflammasome.

The mechanistic target of rapamycin (mTOR) is a key regulator of cellular metabolism, catabolism, immune response, autophagy, survival, proliferation, and migration, thereby maintaining cellular homeostasis [20]. This encompasses responses to food intake, homeostasis, and the pathogenesis of disease [54]. Furthermore, mTOR plays a pivotal role in mediating the process of autophagy [55]. Prior research has demonstrated a robust correlation between the mammalian target of rapamycin (mTOR) and diabetic kidney disease. In this context, kidney cells rely on basal autophagy to survive and maintain kidney integrity [56]. The present study demonstrated that the phosphorylation level of mTOR decreased significantly following the inhibition of the NLRP3 inflammasome. Furthermore, the use of the mTOR activator MHY1485 increased protein expression of p-mTOR and the concomitant inhibition of autophagy-related protein levels following pretreatment with MHY1485. Additionally, the upregulation of pyroptosis-related protein in bovine kidney cells induced by Cd exposure was exacerbated. These findings suggest the existence of an interaction between NLRP3 and mTOR, whereby mTOR-mediated autophagy may exert a protective role in renal cell pyroptosis activated by Cd exposure. Consequently, we proceeded to utilize mTOR inhibitors to ascertain whether such an association existed. CQ is used in research because chloroquine increases the pH of lysosomes, inactivating acidic hydrolases in lysosomes, thereby inhibiting the fusion and degradation of autophagolysosomes in cells. Chloroquine-treated cells result in aggregation of LC3B, where the observed change in LC3B represents only a change in the number of autophagosomes [57]. However, a simple increase in LC3 does not represent an increase in autophagy, so it is compared with a protein treated with an inhibitor to reflect whether the autophagic flow is blocked. Following pretreatment with RAPA, an inhibitor of mTOR, and CQ, we observed an increase in autophagosomes and confirmed the activation of autophagy. This was performed to explore the effect of Cd exposure on inflammatory injury in mouse kidney and bovine kidney cells. The use of RAPA was found to result in the activation of autophagy, as evidenced by the upregulation of autophagy-related proteins and the increased presence of autophagosomes. Additionally, the expression of pyroptosis-related genes was observed to be downregulated, which contributed to the attenuation of Cd exposure-induced inflammatory damage in vivo and in vitro. As previous studies, pyroptosis is regulated by autophagy, and autophagy promotes the inhibition of NLRP3 inflammasome activation and the reduction in pro-inflammatory factor release. Furthermore, autophagy is promoted by inhibiting mTOR phosphorylation, which in turn reduces pyroptosis [58,59,60]. However, previous studies have suggested that the dose of the toxin will increase the secretion of the inflammatory factor IL-1β by activating macroautophagy at a cut-off value, aggravating tissue inflammatory damage [61,62]. Consequently, it is imperative that our research delves more profoundly and explicitly into the effects of autophagy on tissues and cells.

In conclusion, the activation of NLRP3-mediated pyroptosis was observed at selected Cd-stimulated concentrations, accompanied by an increase in the secretion of related inflammatory factors in tissues and cells. This resulted in the promotion of autophagy, and the use of mTOR inhibitors was found to alleviate pyroptosis, thereby improving the damage caused by inflammation. The findings of this study provide a theoretical basis for understanding the nephrotoxic mechanism of Cd. Furthermore, the NLRP3 and mTOR pathways represent potential therapeutic targets for the prevention and treatment of Cd poisoning (Figure 7).

## 4. Materials and Methods

### 4.1. Experimental Animals and Treatments

Thirty male Kunming mice were purchased from the Department of Laboratory Animals, Kunming Medical University (Kunming, China). Animal experiments were performed in accordance with the Guide for the Management and Use of Laboratory Animal Husbandry published by the National Institutes of Health (Bethesda, MD, USA). In accordance with the requirements of the Guide for the Management of Laboratory Animal Feeding, rodent houses, cages, and water jugs were completely sterilized before testing. Tests were performed after 1 week of acclimation feeding. In the subsequent experiments, the mice were divided into 3 groups of 10 mice each according to their average body weight. The Cd-exposed group was given water ad libitum, and the mammalian target of rapamycin (mTOR) inhibitor rapamycin (RAPA) was injected intraperitoneally every 2 days at 2 intervals for 28 consecutive days. The groups were divided as follows: (1) control group; (2) intraperitoneal injection of normal saline and with free-drink 20 μM CdCl_2_ group [63]; (3) intraperitoneal injection of 5 mg/kg BW. RAPA and with free-drink 20 μM CdCl_2_ group (Figure 8). RAPA (MedChemExpress, Monmouth Junction, NJ, USA) was dissolved in dimethyl sulfoxide (Solarbio, Beijing, China). The stock solution was stored in the dark at −80 °C and diluted to the concentration used with saline.

### 4.2. Determination of Biochemical Indices of Serum Renal Function in Mice

Serum CREA and BUN levels were measured according to the instructions of the kit developed by Nanjing Jiancheng.

### 4.3. Hematoxylin and Eosin Staining and Tissue ROS Staining

Mouse kidney tissues were fixed in 4% paraformaldehyde (Servicebio, Wuhan, China) for more than 24 h, and the appropriate size was placed in 1.5 mL centrifuge tubes at −80 °C for fixed storage in Wuhan Servicebio Company for each group of test animals. The pathological and histological changes in mouse kidneys were observed using a light microscope with a classical field of view image.

### 4.4. Cell Culture and Treatments

Madin-Darby Bovine Kidney (MDBK) cells were stored in our lab and cultured in RPMI-1640 (Gibco, New York, NY, USA) containing 10% fetal bovine serum (cell-box, Chang Sha, China) and triple antibiotic (Gibco, New York, NY, USA). Cells in good growth condition were stimulated by adding the appropriate reagents when the density reached 50–80% in different well plates. MDBK cells were exposed to preselected concentrations of CdCl_2_ (0, 10, 20, 40, and 80 μM) for 12 h. In the drug-stimulation assay, cells were pretreated with CY-09 (10 μM), MHY01485 (10 μM), and RAPA (50 nM) 4 h prior to Cd staining (20 μM).

### 4.5. Cell Viability Analysis

The cytotoxicity of different concentrations of CdCl_2_ (0, 10, 20, 40, and 80 μM) on MDBK cells was detected using Cell Counting Kit-8 (Bioss, Beijing, China). The absorbance at 450 nm of each well was detected by a multifunctional enzyme marker (Feyond-A300, Hangzhou, China), and the data were calculated.

### 4.6. Cellular Lactate Dehydrogenase Activity Assay

Cell culture supernatants at the end of stimulation were collected and centrifuged (Cence, Changsha, China) at 12,000 rpm for 15 min at 4 °C. Lactate dehydrogenase (LDH) activity was determined using the LDH cytotoxicity assay kit (Jian Cheng, Nanjing, China).

### 4.7. Extraction of Total RNA and Real-Time Quantitative Reverse Transcription PCR (RT-qPCR)

Cells were lysed using RNAiso Plus (Takara, Kyoto, Japan), and mouse kidney tissue RNA and cellular RNA completed by staining stimulation were extracted with reagents such as chloroform, isopropanol, absolute ethanol, etc., by steps such as shaking and centrifugation according to the manufacturer’s protocol. Then, 1% accounting gel electrophoresis to verify the integrity of RNA, and RNA was reverse transcribed to cDNA using the EasyScript^®^ One-Step gDNA Removal and cDNA (Takara, Kyoto, Japan) Synthesis SuperMix. Quantitative detection of mRNA was performed using a real-time PCR instrument (qTOWER3G, Shanghai, China) with a program set at 94 °C/5 min, 94 °C/30 s→60 °C/30 s × 35 cycles, and was performed by NCBI (National Center for Biotechnology Information) and primer3 Input 4.0 to design the following primers (Table 1). The detection was performed with MCE’s (MedChemExpress, NJ, USA) qPCR Master Mix YBR Green qPCR Master Mix.

### 4.8. Western Blotting

Adherent cells were lysed with RIPA protein lysate (Solarbio, Beijing, China), and their concentrations were determined and normalized using BCA protein quantification kit (Yaenzyme, Shanghai, China). The extracted protein was added to the sample buffer in equal proportions, SDS-PAGE electrophoresis solution was prepared for electrophoresis under bio-rad electrophoresis-transfer device and PVDF membrane for transfer, and then the membrane was placed in the commercial rapid-blocking solution (Yazyme, Shanghai, China) for 15 min, and then the primary antibody was incubated at 4 °C overnight. The following primary antibodies were used: β-actin (rabbit, Bioss Cat#bs-0061r), ATG5 (rabbit, Abcam Cat#ab228668, Cambridge, UK), LC3B (rabbit, Abcam Cat#ab48394), Nucleoprin p62 (rabbit, Abcam Cat#ab96134), mTOR (rabbit, Abcam Cat#ab32028), mTOR (phospho S2448) (rabbit, Abcam Cat#ab109268), NLRP3 (rabbit, Wanleibio, Cat#WL-02635, Shanghai, China), N-GSDMD (rabbit, ABclonal Cat#A-20728, Wooburn, MA, USA), C-caspase (rabbit, ABclonal Cat#A-2156). The incubation was completed with TBST rinsing for 3–5 times × 10 min, and the secondary antibody was incubated at room temperature for 1–2 h after aspiration. The secondary antibodies used were Goat 506 Anti-Rabbit IgG H&L (Bioss, bs-80295G-HRP) and Goat Anti-Mouse IgG H&L (Bioss, bs-0296G-HRP).

### 4.9. Flow Cytometry

MDBK cells exhibiting optimal growth characteristics were inoculated in six-well plates and treated with varying concentrations of Cd dye for 12 h. Following this, the cells were washed three times with PBS, digested using EDTA-free trypsin, and transferred to centrifuge tubes. The cells were resuspended by adding 1 mL of DCFH-DA (Beyotime, Shanghai, China) dye working solution configured in serum-free medium to the cell precipitate and incubated at 37 °C without light for 30 min. Thereafter, the cells were washed with a serum-free medium once, and the intracellular ROS level was detected by flow cytometry.

### 4.10. Transmission Electron Microscopy

The cultured cells were discarded from the medium, gently scraped off with a cell scraper (trying not to digest enzymatically), collected and enriched by low-speed centrifugation, and the supernatant was discarded. Then, the cells were resuspended with room-temperature electron microscope fixative (Scientist, Sichuan, China), transferred together with the fixative to a 1.5 mL sharp-bottom EP tube, and immediately centrifuged continuously at 10,000 rpm for 10–12 min (the cells must be centrifuged as soon as possible within 1 min, as they will not be tightly separated for a long period, and then they will be easily dispersed and lost in the subsequent preparation of samples). After centrifugation, the cell precipitate should be no more than half the size of a green bean. The final steps were to carefully discard the supernatant, slowly add new electron microscope fixative along the wall of the tube without impacting the cell mass, leave it for 30 min, and then transfer it to 4 °C for storage and entrust it to Sichuan Scientist Biotechnology Co., Ltd. (Scientist, Sichuan, China) for shooting.

### 4.11. Statistical Analysis

GraphPad Prism version 8.0 (GraphPad software) was used for statistical analysis. All data were presented as mean ± SD, one-way analysis of variance followed by Tukey’s post-tests were applied for analyzing the differences among multiple groups. * *p* < 0.05, ** *p* < 0.01 and *** *p* < 0.0001 were considered statistically significant versus the control group and NC. # *p* < 0.05, ## *p* < 0.01 and ### *p* < 0.0001 were considered statistically significant versus the CdCl_2_ group. All experiments were repeated three times.

## Figures and Tables

**Figure 1 ijms-26-02589-f001:**
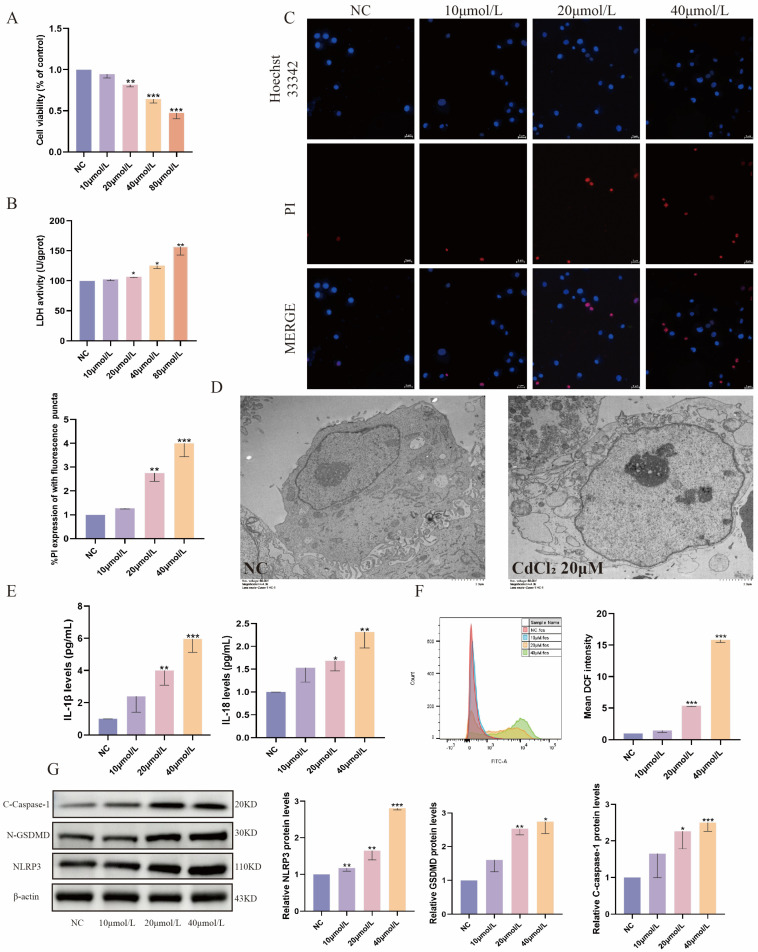
Cadmium exposure-induced pyroptosis in MDBK cells. (**A**) Cell viability; (**B**) LDH activity; (**C**) Hoechst/PI staining; (**D**) transmission electron microscopy scanning; (**E**) the contents of *IL−1β* and *IL−18* in the culture medium; (**F**) ROS levels in MDBK; (**G**) the levels of pyroptosis-related proteins. Data are presented as mean ± SD (*n* = 3). * *p* < 0.05, ** *p* < 0.01, and *** *p* < 0.0001 were considered statistically significant vs. the NC.

**Figure 2 ijms-26-02589-f002:**
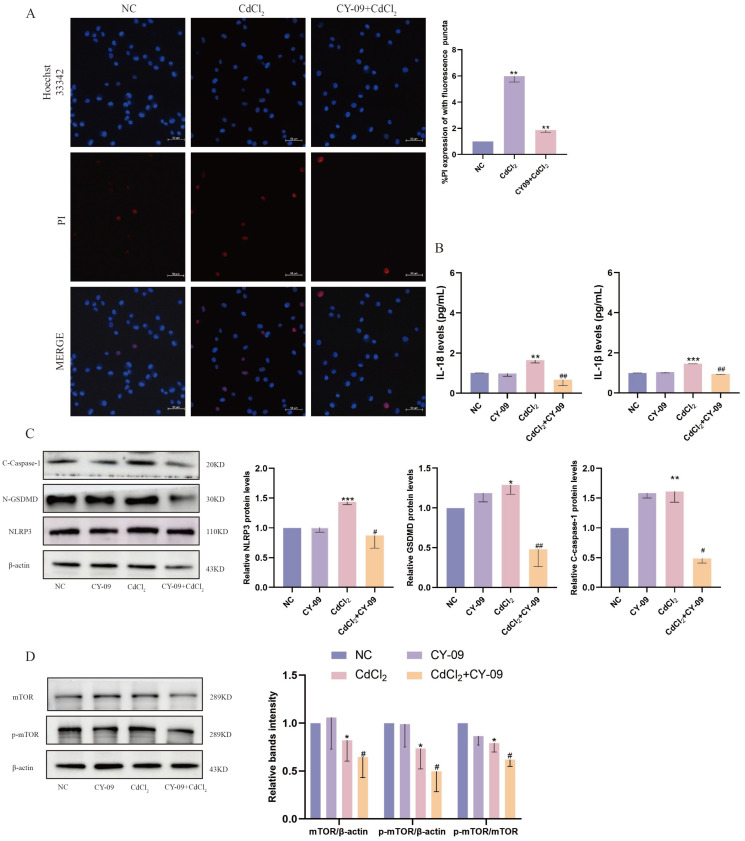
Inhibition of NLRP3 alleviates cadmium-induced pyroptosis in MDBK cells. (**A**) Hoechst/PI staining; (**B**) the contents of *IL−1β* and *IL−18* in the culture medium; (**C**) the levels of pyroptosis-related proteins; (**D**) the phosphorylation of mTOR was detected by Western blotting. Data are presented as mean ± SD (*n* = 3). * *p* < 0.05, ** *p* < 0.01, and *** *p* < 0.0001 were considered statistically significant vs. the NC. # *p* < 0.05, ## *p* < 0.01, were considered statistically significant vs. the CdCl_2_ group.

**Figure 3 ijms-26-02589-f003:**
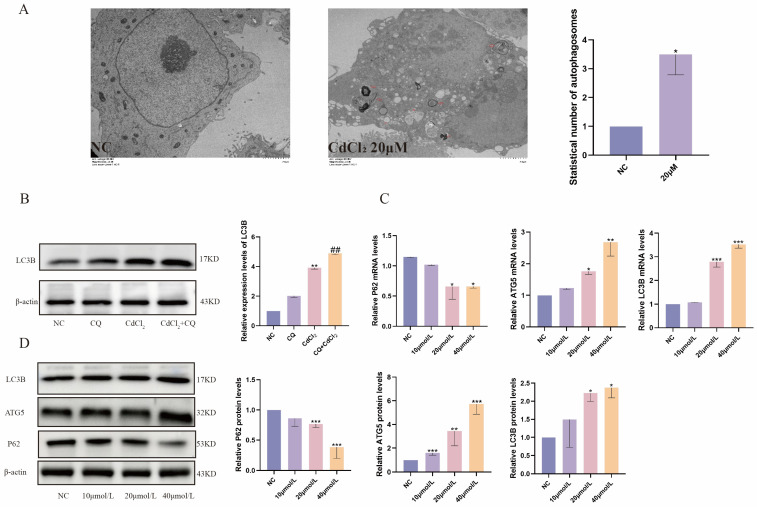
Cadmium exposure promoted autophagy in MDBK cells. (**A**) Transmission electron microscopy scanning. The red arrows in the diagram refer to autophagosomes; (**B**) the expression levels of autophagy-related proteins; (**C**) expressions of autophagy-related genes in mRNA levels; (**D)** the expression of autophagy-related proteins. Data are presented as mean ± SD (*n* = 3). * *p* < 0.05, ** *p* < 0.01, and *** *p* < 0.0001 were considered statistically significant vs. the NC. ## *p* < 0.01, was considered statistically significant vs. the CdCl_2_ group.

**Figure 4 ijms-26-02589-f004:**
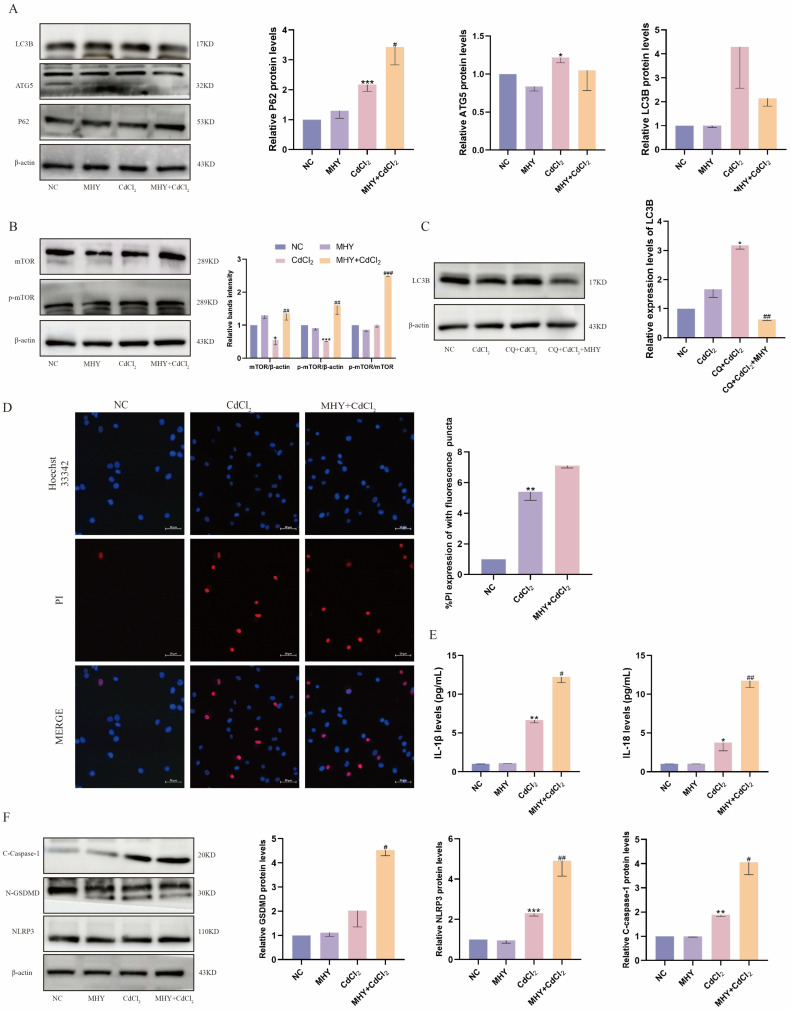
Activation of mTOR exacerbates Cd-induced pyroptosis. (**A**) The expression of autophagy-related proteins; (**B**) the phosphorylation level of mTOR in MDBK; (**C**) the autophagic flow in MDBK cells; (**D**) Hoechst/PI staining; (**E**) the contents of *IL−1β* and *IL−18* in the culture medium; (**F**) the pyroptosis-related protein expression. Data are presented as mean ± SD (*n* = 3). * *p* < 0.05, ** *p* < 0.01, and *** *p* < 0.0001 were considered statistically significant vs. the NC. # *p* < 0.05, ## *p* < 0.01, and ### *p* < 0.0001 were considered statistically significant vs. the CdCl_2_ group.

**Figure 5 ijms-26-02589-f005:**
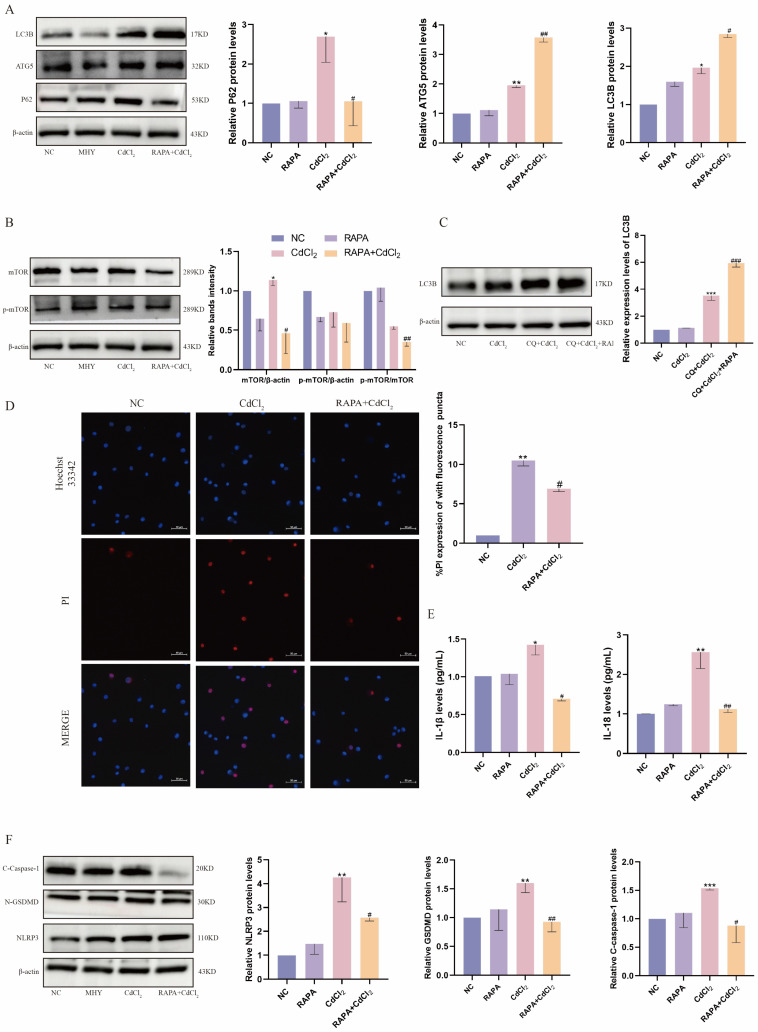
Inhibition of mTOR attenuates Cd-induced pyroptosis in MDBK cells. (**A**) The expression of autophagy-related proteins; (**B**) the phosphorylation level of mTOR in MDBK; (**C**) the autophagic flow in MDBK cells; (**D**) Hoechst/PI staining; (**E**) the contents of *IL−1β* and *IL−18* in the culture medium; (**F**) the pyroptosis-related proteins expression. Data are presented as mean ± SD (*n* = 3). * *p* < 0.05, ** *p* < 0.01, and *** *p* < 0.0001 were considered statistically significant vs. the NC. # *p* < 0.05, ## *p* < 0.01, and ### *p* < 0.0001 were considered statistically significant vs. the CdCl_2_ group.

**Figure 6 ijms-26-02589-f006:**
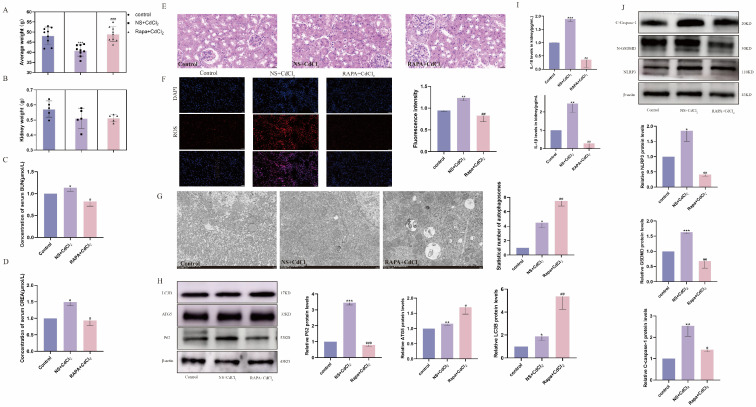
Inhibition of mTOR alleviates Cd-induced pyroptosis in mouse kidney tissue. (**A**) Body weight in mice; (**B**) kidney weight in mice; (**C**) serum BUN level; (**D**) serum CREA level; (**E**) H&E staining, scale bar: 50 μm; (**F**) fluorescence intensity of mouse kidney ROS, scale bar: 20 μm; (**G**) transmission electron microscopy scanning; (**H**) the expression of autophagy-related proteins; (**I**) the contents of *IL−1β* and *IL−18* in the mouse serum; (**J**) the expression of pyroptosis-related proteins. Data are presented as mean ± SD (*n* = 10). * *p* < 0.05, ** *p* < 0.01, and *** *p* < 0.0001 were considered statistically significant vs. the control group. # *p* < 0.05, ## *p* < 0.01, and ### *p* < 0.0001 were considered statistically significant vs. the NS + CdCl_2_ group.

**Figure 7 ijms-26-02589-f007:**
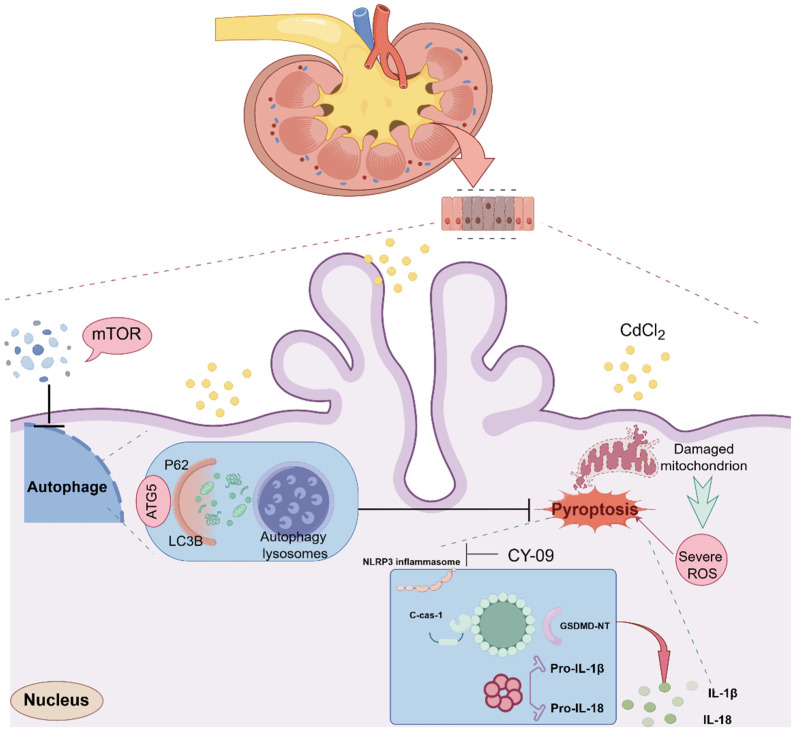
Schematic diagram of mTOR-mediated autophagy in regulating cadmium-induced renal injury via pyrolysis (By Figdraw 2.0).

**Figure 8 ijms-26-02589-f008:**
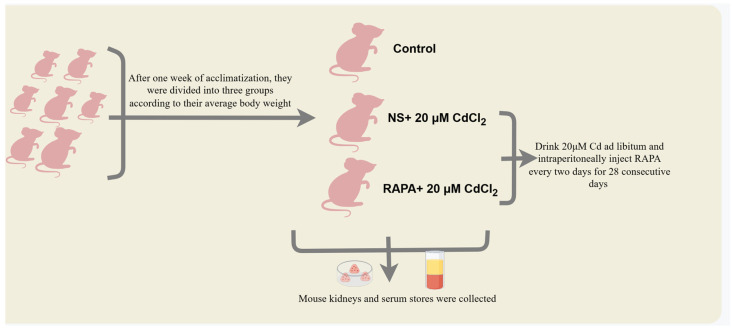
Animal experiment grouping (By Figdraw 2.0).

**Table 1 ijms-26-02589-t001:** RT-qPCR sequence of primer.

Gene	Sequence (5′–3′)	Size
Cattle: *β-actin*:	F: GCAGGAGTACGATGAGTCCG	103
	R: GCAGGAGTACGATGAGTCCG	
*IL-1β*:	F: AAAAATCCCTGGTGCTGGCT	138
	R: CATGCAGAACACCACTTCTCG	
*IL-18*:	F: CCTTTGAGGCATCCAGGACA	116
	R: CACACCACAGGGGAGAAGTG	
Mouse: *β-actin*:	F: CCAACCGTGAGAAGATGAC	220
	R: AGGCATACAGGGACAGCACA	
*IL-1β*:	F: AACCTGCTGGTGTGTGACGTTC	128
	R: CAGCACGAGGCTTTTTTGTTGT	
*IL-18*:	F: GTGAACCCCAGACCAGACTG	114
	R: CCTGGAACACGTTTCTGAAAGA	

## Data Availability

The original contributions presented in this study are included in the article. Further inquiries can be directed to the corresponding authors.

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
