# Peer review of "mTOR-Mediated Autophagy Regulates Cadmium-Induced Kidney Injury via Pyroptosis"

_ijms, 2025, doi:10.3390/ijms26062589_

Round 1
Reviewer 1 Report
Comments and Suggestions for Authors
Aim of this study is to examine if Cd induced injury of kidney through pyroptosis can be diminished by mTOR mediated autophagy. Effect of Cd is evaluated in vitro (MDBK cells) and in vivo in Kumming mice (control group, 20mM CdCl2 group and 5mg/kg b.w.+ 20mM CdCl2 group). Cd solely increased LDH (violates integrity of kidney cell membrane), expression of pyroptosis-associated proteins (NLRP3/GSDMD/casp-1, ATG5/LC3B), and promoted the release of inflammatory cytokines (IL-1b, IL-18). After administration of NLRP3 inhibitor (CY-09) Cd induced pyroptosis was attenuated. It has been shown that mTOR inhibitor (rapamicyn) down regulated the expression of pyroptosis-related proteins,which indicates a protective effect of mTOR-mediated autophagy n NLRP3 inflammasome-dependent kidney injury after Cd exposure.
The Manuscript is well written and it can be seen that a lot of effort was invested in the implementation and analysis of such a large number of results, with the aim of examining the mechanisms. Some minor changes should be included:
1. Introduction section has a lot of sentences which begin with „Cd…“so please rephrase them.
2. Please explain why in in vivo set of experiments it is examined Cd concentration od 20mM solely? Why not 40mM as well?
3. Line 60-62: please add reference in brackets.
4. In Fig. 2B maybe it would be better to set the y-axis to 6 pg/ml for easier comparison with the cytokine level in Fig 1E.
5. In Fig. 2C on x-axis it does not say what concentration of Cd was used, is it 20mM?
6. Section 3.4. refers to Fig. 4A-F, not Fig. 3A-F, so please correct it.
7. I suggest to present graphics with the CQ effect (Fig. 3B and 3C) as a separate Figure.
8. Line 346: "However,..." should be preceded by a period.
Author Response
1.Introduction section has a lot of sentences which begin with „Cd…“so please rephrase them.
Response : Many thanks for your positive and constructive comments and suggestions. We have revised it in the manuscript (Introduction , lines 29-31).
2.Please explain why in in vivo set of experiments it is examined Cd concentration od 20mM solely? Why not 40mM as well?
Response : Many thanks for your positive and constructive comments and suggestions. Combined with the literature Exposure to low-dose cadmium induces testicular ferroptosis[1], we applied 20mM(lines 100).
3.Line 60-62: please add reference in brackets.
Response : Many thanks for your positive and constructive comments and suggestions.We have added citations to the manuscript (lines 67).
4.In Fig. 2B maybe it would be better to set the y-axis to 6 pg/ml for easier comparison with the cytokine level in Fig 1E.
Response : Many thanks for your positive and constructive comments and suggestions. We have revised it in manuscript Fig. 2B.
5.In Fig. 2C on x-axis it does not say what concentration of Cd was used, is it 20mM?
Response : Thanks very much for your positive and constructive comments and suggestions. Yes, the concentrations we screened out in Result One were used for subsequent experiments.
6.Section 3.4. refers to Fig. 4A-F, not Fig. 3A-F, so please correct it.
Response : Thanks very much for pointing it out, We have corrected it in the manuscript.
7.I suggest to present graphics with the CQ effect (Fig. 3B and 3C) as a separate Figure.
Response : Many thanks for your positive and constructive comments and suggestions. We have shown it separately in the manuscript.
8.Line 346: "However,..." should be preceded by a period.
Response : Thanks very much for pointing it out, We have corrected it in the manuscript.

Reviewer 2 Report
Comments and Suggestions for Authors
Reviewer(s)' Comments to Author:
This paper presented by Hu et al. explored the role of mTOR in Cd-induced kidney injury. And the data showed that mTOR-mediated autophagy has a significant protective effect on NLRP3 inflammasome-dependent kidney cell pyroptosis, which provides a new insight into the prevention and treatment of Cd poisoning. Overall, this is an interesting and generally well-done study. Most of the conclusions are adequately supported by the data shown. There are several minor questions that should be addressed.
Specific comments:
- The description of the results in the Abstract section is not clear enough, and it is necessary to highly summarize and generalize the main experimental results.
- The hazards of heavy metal cadmium exposure to animal husbandry need to be described in the Introduction section, such as the prevalence of cadmium poisoning in animals and the economic damage it causes.
- It would be better if a flowchart could be used to display the grouping and processing of animal experiments.
- The product size of primers in the qPCR experiment should be provided.
- The results of PI staining should be statistically analyzed.
- The number of autophagosomes and autolysosomes in TEM should be statistically analyzed.
- CQ is an autophagy inhibitor that can inhibit the fusion of autophagosomes and lysosomes. What is the purpose of using CQ in this study? The authors need to explain it in the Results and Discussion sections. Besides, the vertical axis name in Figure 3B/4C/5C should refer to the relative expression level of LC3B. Please correct it.
- The reference number of the figure in 3.4 is incorrect, not Figure 3A-F, but Figure 4A-F. Please revise it.
- The layout of Figure 6 is too long, please modify it to a suitable size.
- Figure 7 needs to be cited in the main text.
- Although English writing is generally good, there are still some grammar errors that need to be checked throughout the text.

- Although English writing is generally good, there are still some grammar errors that need to be checked throughout the text.
Author Response
Comment 1. The description of the results in the Abstract section is not clear enough, and it is necessary to highly summarize and generalize the main experimental results.
Response: Thanks very much for your positive and constructive comments and suggestions. We've made streamlined changes to the abstract results section (lines 19-20).
Comment 2. The hazards of heavy metal cadmium exposure to animal husbandry need to be described in the Introduction section, such as the prevalence of cadmium poisoning in animals and the economic damage it causes.
Response: Many thanks for your valuable comments. We have already described the hazards of heavy metal cadmium exposure to animal husbandry in the introduction section (lines 36-42).
Comment 3. It would be better if a flowchart could be used to display the grouping and processing of animal experiments.
Response: Thanks very much for your valuable comments. We have added a flowchart for grouping animal experiments to the manuscript.
Comment 4. The product size of primers in the qPCR experiment should be provided.
Response: Many thanks for your positive and constructive comments and suggestions. We have added primer product sizes to Table 1.
Comment 5. The results of PI staining should be statistically analyzed.
Response: Many thanks for your positive and constructive comments and suggestions, and we are very sorry for the unclear description. We have added statistical analysis of PI staining to the relevant fig of the manuscript.
Comment 6. The number of autophagosomes and autolysosomes in TEM should be statistically analyzed.
Response: Many thanks for your kind suggestions. We have added statistical analysis of autophagosomes to the manuscript fig3.
Comment 7. CQ is an autophagy inhibitor that can inhibit the fusion of autophagosomes and lysosomes. What is the purpose of using CQ in this study? The authors need to explain it in the Results and Discussion sections. Besides, the vertical axis name in Figure 3B/4C/5C should refer to the relative expression level of LC3B. Please correct it.
Response: Thank you very much for your questions and suggestions. CQ is used in research because chloroquine increases the pH of lysosomes, inactivating acidic hydrolases in lysosomes, thereby inhibiting the fusion and degradation of autophagolysosomes in cells. Chloroquine-treated cells result in aggregation of LC3B, where the observed change in LC3B represents only a change in the number of autophagosomes[1]. However, a simple increase in LC3 does not represent an increase in autophagy, so it is compared with a protein treated with an inhibitor to reflect whether the autophagic flow is blocked. We have changed this in Figures 3B/4C/5C of the manuscript.
Comment 8. The reference number of the figure in 3.4 is incorrect, not Figure 3A-F, but Figure 4A-F. Please revise it.
Response: Thanks very much for pointing it out, We have corrected it in the manuscript.
Comment 9. The layout of Figure 6 is too long, please modify it to a suitable size.
Response: Thank you very much for your suggestion. We have modified the layout dimensions of fig6 in the manuscript.
Comment 10. Figure 7 needs to be cited in the main text.
Response: Thank you very much for your suggestion. We have already cited fig7 in the manuscript.
Comment 11. Although English writing is generally good, there are still some grammar errors that need to be checked throughout the text.
Response: Thank you very much for your guidance and advice. We have checked the grammar in the manuscript and corrected it.
